# Multilocus sequence typing of *Candida albicans* oral isolates reveals high genetic relatedness of mother-child dyads in early life

**Naemah Alkhars**[1,2], **Nisreen Al Jallad**[3], **Tong Tong Wu**[4], **Jin Xiao**[3]*

1 Department of General Dental Practice, College of Dentistry, Health Science Center, Kuwait University, Safat, Kuwait, 2 Translational Biomedical Science Program, Clinical and Translational Science Institute, University of Rochester School of Medicine and Dentistry, Rochester, New York, United States of America, 3 Eastman Institute for Oral Health, University of Rochester Medical Center, Rochester, New York, United States of America, 4 Department of Biostatistics and Computational Biology, University of Rochester Medical Center, Rochester, New York, United States of America

* jin_xiao@urmc.rochester.edu

**Data Availability Statement:** Sequence reads were deposited and are available from the NCBI Sequence Read Archive (SRA; Bioproject number PRJNA926612). Additional isolate sequences and

## Abstract

*Candida albicans* is a pathogenic fungus recently recognized for its role in severe early childhood caries development (S-ECC). *C. albicans* oral colonization begins at birth, but the extent of the mother's involvement in yeast transmission to their children is unclear, therefore, this study used a prospective mother-infant cohort to investigate the maternal contribution of *C. albicans* oral colonization in early life. Oral samples were collected from 160 mother-child dyads during pregnancy and from birth to two years of life. We used whole-genome sequencing to obtain the genetic information of *C. albicans* isolates and examined the genetic relatedness of *C. albicans* between mothers and their children using Multilocus Sequence Typing. Multivariate statistical methods were used to identify factors associated with *C. albicans'* acquisition (horizontal and vertical transmissions). Overall, 227 *C. albicans* oral isolates were obtained from 93 (58.1%) of mother-child pairs. eBURST analysis revealed 16 clonal complexes, and UPGMA analysis identified 6 clades, with clade 1 being the most populated 124 isolates (54.6%). Significantly, 94% of mothers and children with oral *C. albicans* had highly genetically related strains, highlighting a strong maternal influence on children's *C. albicans* acquisition. Although factors such as race, ethnicity, delivery method, and feeding behaviors did not show a significant association with *C. albicans* vertical transmission, the mother's oral hygiene status reflected by plaque index (PI) emerged as a significant factor; Mothers with higher dental plaque accumulation (PI >=2) had a significantly increased risk of vertically transmitting *C. albicans* to their infants [odds ratio (95% confidence interval) of 8.02 (1.21, 53.24), p=0.03]. Furthermore, Black infants and those who attended daycare had an elevated risk of acquiring *C. albicans* through horizontal transmission (p <0.01). These findings highlight the substantial role of maternal transmission in the oral acquisition of *C. albicans* during early life. Incorporating screening for maternal fungal oral carriage and implementing oral health education programs during the perinatal stage may prove valuable in preventing fungal transmission in early infancy.

metadata were obtained from the PubMLST database (https://pubmlst.org). All other relevant data are within the manuscript and its Supporting Information files.

**Funding:** JX; grants K23DE027412 and R01DE031025 from the National Institute of Dental and Craniofacial Research. https://www.nidcr.nih.gov/ The funders had no role in study design, data collection and analysis, decision to publish, or preparation of the manuscript.

**Competing interests:** The authors have declared that no competing interests exist.

## Introduction

Fungi are diverse and abundant microbial colonizers of human bodies, forming an important part of the mycobiome. Alterations in the mycobiome of specific niches, termed dysbiosis, have been implicated in a range of human diseases [1]. Krom et al. have suggested that fungi may act as keystone species in establishing and maintaining healthy oral ecosystems [2]. The interactions between fungal species and humans can range from mutualistic to parasitic, with the dynamic interplay between them being crucial for maintaining host health [1]. *Candida albicans* is the most commonly detected fungal organism on human mucosal surfaces [3], and it has been shown to colonize the oral cavity as early as a few hours after birth [4, 5]. This early colonization poses a significant health risk to immunocompromised infants, particularly those with low birth weights [6]. In pediatric medicine, contagious colonization of newborn infants by *C. albicans* has been recognized, and the transmission has been reported to be vertical and horizontal [7]. Moreover, *C. albicans* can cause a range of infections in humans, from mucosal infections to systemic infections [8].

*C. albicans* is a diploid fungal species with significant genetic heterozygosity. Its genome consists of eight pairs of chromosome homologs, ranging in size from 0.95 to 3.3 Mb, totaling up to 16 Mb [9]. This heterozygosity has implications for the study of this organism, including its phenotyping and genotyping. S1 Fig depicts the phylogenetic position of *C. albicans* within the fungal kingdom.

Phenotyping and genotyping of *Candida* strains are crucial methods for investigating noso-comial candidiasis. These methods can identify outbreak-related strains, determine the origins of infection, and track the transmission of strains. Additionally, genotyping can help evaluate the diversity of isolates within a carrier and investigate recurrent infections to identify particularly virulent strains, if any. Furthermore, tracking the emergence of drug-resistant strains is vital in guiding the development of new antifungal therapies [10]. The species' population structure and diversity can also be studied using genotyping methods. For instance, Multilocus Sequence Typing (MLST) is a useful method for identifying the genetic diversity of isolates within a population and investigating evolutionary relationships among them [11].

In epidemiological studies, the typing methods used to characterize *C. albicans* isolates vary considerably. Molecular characterization methods are more discriminatory and accurate than phenotypic methods and are therefore preferred. There are two major classes of *C. albicans* typing methods: non-DNA-based and DNA-based techniques [11]. Non-DNA-based methods include protein fingerprint comparisons and Multilocus Enzyme Electrophoresis (MLEE), while DNA-based methods directly analyze polymorphisms within various DNA markers. DNA-based techniques are classified into conventional typing methods and exact DNA-based techniques. The conventional typing methods include electrophoretic karyotyping (EK), restriction enzyme analysis (REA), and random amplified polymorphic DNA (RAPD). Furthermore, exact DNA-based techniques generate highly reproducible typing data, such as MLST [11], which has been used successfully in the population genetics and molecular phylogeny studies of *C. albicans* [12]. Pulsed Field Gel Electrophoresis (PFGE), Ca3 fingerprinting, and MLST are highly discriminatory and accurate typing methods that outperform phenotypic techniques [13]. Therefore, it is feasible to lead a *C. albicans* relatedness analysis using the MLST technique [14].

*C. albicans* is a ubiquitous fungal organism that commonly colonizes the oral cavity of healthy individuals. The prevalence of *C. albicans*' carriage varies depending on multiple factors, including age, gender, diet, geographic location, socioeconomic status, immunosuppression, and medication use [15]. Studies have reported that the oral carriage of *C. albicans* has been detected in 46% of healthy infants and 69% of adults [8, 16]. However, infants are more

susceptible to oral candidiasis, with up to 37% of infants experiencing this opportunistic fungal infection, especially until six months of age [17]. Elderly individuals who wear dentures are also at increased risk of developing oral candidiasis, with up to 54% of denture wearers experiencing this condition [17]. Pregnant and postpartum women are also more likely to experience oral candidiasis and *C. albicans* colonization, possibly due to hormonal changes that affect the oral microbiome [18, 19]. Moreover, *C. albicans* is also known to cause infections in other body sites, including the genital area. Vulvovaginal candidiasis (VVC) is a common fungal infection that affects up to 75% of women at some point in their lives [20]. Pregnant women, especially in their third trimester, are at higher risk of VVC due to increased estrogen levels in the vaginal mucosa that promote yeast adhesion and penetration [21, 22].

*C. albicans* is a polymorphic fungus, which can switch between different morphological forms, such as yeast, hyphae, and pseudohyphae, depending on the environmental conditions [23]. This polymorphic nature is considered a key virulence factor that plays a critical role in the transition from commensalism to parasitism [1]. The ability of *C. albicans* to switch between these forms is thought to facilitate its survival and persistence within the host. Recent studies have shown that the presence of *C. albicans* can contribute to the development of dental caries by enhancing the cariogenicity of *Streptococcus mutans*, a bacterium commonly associated with dental caries [24, 25]. This may occur through the production of organic acids such as lactic acid by both *C. albicans* and *S. mutans*. *C. albicans* is also known to be capable of fermenting carbohydrates in the diet, producing acids that can dissolve away the mineralized tooth structure, leading to the development of dental caries [26, 27]. Epidemiological data have suggested that *C. albicans* is involved in the pathogenesis of dental caries [28]. Studies have reported that *C. albicans* can be detected in the saliva and dental plaque of individuals with dental caries and that its concentration is higher in those with active caries than in those without [24, 29, 30]. Furthermore, *C. albicans* has been shown to form biofilms on the surface of teeth, which can promote the retention of other microorganisms and contribute to the development of dental caries [25].

Recent research has revealed that the significance of *C. albicans* as a potent caries microbe has often been underestimated compared to the well-known cariogenic *S. mutans* [31]. While maternal transmission of *S. mutans* has been extensively studied and documented as a method by which children are initially colonized with this bacterium [32], transmission patterns of *C. albicans* in children at high risk of severe early childhood caries (S-ECC) have not yet been investigated. To address this gap in knowledge, a study was conducted to identify and genotype oral *C. albicans* isolates from mother-child dyads to determine the possibility of vertical transmission in early life. The study also aimed to identify the factors associated with the vertical and horizontal transmission of *C. albicans*, which could help design measures to limit and prevent the transmission of this pathogenic fungi, possibly reducing the burden of S-ECC experience.

## Material and methods

### Study population and *Candida* isolates

Pregnant women and their children were obtained from a birth cohort study that studied the association between early-life oral *Candida* colonization and the onset of dental caries in children [33]. The study protocol was approved by the University of Rochester Research Subject Review Board (#67191). Written informed consent was obtained from pregnant women to participate in the study and allow the review of their medical records. For children, their legal guardians reviewed and signed a written permission form authorizing their participation and the review of their medical records. The participants were recruited from patients who visited

the University of Rochester Highland Family Medicine (HFM) or Eastman Institute for Oral Health (EIOH) Perinatal Dental Clinic between November 2017 and August 2020. The study was fully completed in May 2022. Self-reported questionnaires were administered to gather information on the demographic, socioeconomic, oral behavior, medical, and medication backgrounds of the mothers and children. These questionnaires were then cross-checked with the subjects' electronic medical records. The authors had access to identifiable participant information during and after the data collection process. To determine the required sample size, a calculation was performed based on the estimated proportion of 60% for the occurrence of identical or highly genetically related *C. albicans* strains between mothers and children. This proportion was compared with the range of 14% to 41% reported in the literature for the general population [7, 34]. The average of these reported proportions (35%) was used as the null proportion in the calculation. A one-sided z-test with an alpha level of 0.05 was employed to achieve 80% power. Based on these considerations, a total of 30 mother-child pairs were required for the study. The study employed a comprehensive protocol using established techniques for clinical examination and sample collection [35, 36]. Clinical isolates of *Candida* species were obtained from the saliva and dental plaque of pregnant women and their children, as described in S1 Appendix. These isolates were identified based on their specific color after being grown on BBL™ CHROMagar™ Candida (BD, Sparks, MD, USA). Two isolates per sample were selected and stored in a sterilized 1.5 ml Eppendorf tube and kept frozen in a -80˚C freezer for future use. For our current study, one isolate was used for subsequent analysis. The study comprised eight visits: prenatal (during the mother's third trimester) for the mothers and 1, 2, 4, 6, 12, 18, and 24 months after delivery, for the infants. The initial/baseline visit for infants typically occurred at 1 month of age. However, for infants who missed the 1-month visit, their initial visit is considered to be the first one when they infant were enrolled in the study. In the case of mothers (n=51), only one isolate was analyzed. Among the children (n=78), the number of isolates varied from one subject to another. For each study visit, when each child tested positive for *C. albicans*, one isolate was selected for MLST analysis. S1 Table details the number and source of each isolate at different study visit times.

## Inclusion and exclusion criteria

Third-trimester pregnant women (beyond 28 gestational weeks), 18 years of age or older, and eligible for New York State-supported medical insurance based on their income level (≤138% of the federal poverty line) were included in the study. Mothers and infants who had received oral and/or systemic antifungal therapy within 90 days of the initial study visit or had severe systemic medical conditions (such as human immunodeficiency virus infection) that increased their susceptibility to yeast infections were excluded from the study. Infants included in the study were born to the participating mothers, with the exception of those who met any of the following exclusion criteria: 1) being born prematurely (before 37 weeks of gestation), 2) having a low birth weight (less than 2,500 grams), 3) having Down syndrome, 4) having orofacial deformities (such as cleft lip, cleft palate, or oral-pharyngeal mass), or 5) having received oral and/or systemic antifungal treatment prior to the initial study visit.

## Oral examination and data/sample collection

Comprehensive oral examination and data/sample collection occurred at all study visits. A comprehensive oral examination (caries score, plaque index, and oral candidiasis) and oral sample collection (saliva and plaque) were performed by 1 of 3 calibrated dentists in a dedicated examination room at the University of Rochester clinics, using standard dental examination equipment, materials, and supplies, under portable lighting, using methods detailed in S1

Appendix. To ensure consistency and reliability of the evaluated criteria, inter- and intra-examiner agreement was determined by κ statistics and exceeded 83% at the calibration.

## DNA extraction, illumina library preparation, and whole genome sequencing

Cells from -80˚C stock cultures were streaked on Yeast Peptones Dextrose agar (YPD) and then incubated for 48 hours at 37˚C. One colony was randomly selected from each dish for DNA extraction, and overnight cultures were prepared. The clinical isolates' whole genome DNA (gDNA) was extracted using the MasterPure™ kit (Lucigen Corp, Middleton, WI, USA), following the manufacturer's instructions. Next, the Nextera XT kit (Illumina, Inc., San Diego, CA, USA) was used to construct libraries with 3 ng of gDNA as input. The Fragment Analyzer and Qubit were used to measure the fragment size profiles and quantify the libraries, respectively. The libraries were standardized to 1.75 nM and sequenced on a NovaSeq 6000 using an SP flow cell with 150-bp paired-end reads. Sequence reads were deposited in the NCBI Sequence Read Archive (SRA: Bioproject number PRJNA926612).

## Genome assembly

TrimGalore (version 0.6.7) [37] was used to remove Nextera adapters, low-quality reads (quality <25), and short reads (length <75). The trimmed and filtered reads were used as input into SPAdes (version 3.15.2) [38] for de-novo genome assembly with default settings.

## MLST of *C. albicans* isolates

MLST was performed by the sequencing results of the internal fragment of seven *C. albicans* housekeeping genes (*AAT1a*, *ACC1*, *ADP1*, *MPIb*, *SYA1*, *VPS13*, and *ZWF1b*), ranging from 373-491 base pairs (bps) and producing a total of 2883 concatenated nucleotide bps (S2 Table) [39]. Samples that were found to be other than *C. albicans* were excluded from the analysis, resulting in a complete MLST analysis of 227 samples (51 mothers and 78 children). Novel and previously known alleles for each gene were identified and given an integer number corresponding to an "allelic profile" using the non-redundant database program on the MLST website (https://pubmlst.org). The combination of the seven distinct allelic profiles for each isolate was then considered a unique diploid sequence type (DST). New allele numbers, including new DSTs, were provided by the curator: Dr. Marie-Elisabeth Bougnoux, and their details are presented in S3 Table.

## Phylogenetic analysis and global population structure of *C. albicans*

To investigate the evolutionary relationships among *C. albicans* isolates, we constructed a dendrogram using the unweighted pair group method with an arithmetic average (UPGMA) analysis of the concatenated MLST sequences [40, 41]. A further 3000 validated isolates archived in the MLST database (https://pubmlst.org) were included in the phylogenetic analysis to explore the evolutionary relationships between the DSTs. The UPGMA trees were generated using MEGA 11 software [42] and displayed using the online tool, Interactive Tree of Life [43]. In addition, we generated a minimum spanning (MS) tree and neighbor-joining (NJ) tree [44] using the online software available at the database (https://online.phyloviz.net/index) [45] and MEGA 11 software [42], respectively. To investigate the evolutionary ancestry patterns among the *C. albicans* isolates, we categorized the fungal population into clonal complexes (CCs). These CCs represent clusters of genetically closely related isolates. The CCs and the founder DSTs, where possible, were obtained using global optimal eBURST (goeBURST) analysis

through the PHYLOViZ 2.0 software [46]. Isolates that shared five out of the seven MLST genes were considered to belong to the same CC [46]. The output of this analysis demonstrated the simplest patterns of descent from the ancestral type for each DST (Fig 3).

## Data and statistical analysis

Children were grouped based on their oral *C. albicans* status, and the characteristics of the two groups (*C. albicans* positive vs. *C. albicans* negative) were compared using the t-test for continuous variables and Pearson's chi-square or Fisher's exact tests for categorical variables.

Categorical variables were presented as frequencies (%), and continuous variables were described using mean and standard deviation. Multiple logistic regression was used to evaluate the factors associated with the vertical and horizontal transmission of *C. albicans* isolated from children (Y/N). The data were analyzed using R Studio v4.2.2. Statistical significance was determined by a p-value < 0.05.

## Results

### Clinical and demographic features

The studied population included 160 mother-child dyads; Table 1 illustrates the demographic, socioeconomic, medical, and oral characteristics of the participating mothers and their children. The results indicated no significant association between the detection status of *C. albicans* and child gender, ethnicity, or birth route (p-value > 0.05). However, Black children had a higher rate of *C. albicans* detection compared to their White or other races counterparts (p-value = 0.007). Furthermore, children diagnosed with oral thrush had a significantly higher rate of *C. albicans* detection (p-value < 0.001). Additionally, children in the *C. albicans* positive group received more antifungal treatment and showed a higher rate of *S. mutans* detection by

**Table 1. Demographic, socioeconomic, medical, and dental background of mother-child dyads.**

| Categories | | | | Child *C. albicans* Status | | |
|---|---|---|---|---|---|---|
| | | | Enrolled n=160 (%) | Negative n=82 (%) | Positive n=78 (%) | p value |
| Gender | Male | | 80 (50) | 40 (49) | 40 (51) | 0.75 |
| Race | Black | | 91 (57) | 37 (45) | 54 (69) | **0.007** |
| | White | | 34 (21) | 21 (26) | 13 (17) | |
| | Other | | 35 (22) | 24 (29) | 11 (14) | |
| Ethnicity | Non-Hispanic | | 136 (85) | 68 (83) | 68 (87) | 0.45 |
| Birth weight (gm) | | | 3313±425 | 3315±389 | 3312±462 | 0.96 |
| Birth route | Vaginal | | 119 (74) | 59 (72) | 60 (77) | 0.47 |
| Siblings (Y) | | | 107 (67) | 54 (66) | 53 (68) | 0.78 |
| Ever breastfeeding (Y) | | | 115 (72) | 63 (77) | 52 (67) | 0.15 |
| Night bottle feeding (2 months) | | | 102 (64) | 44 (54) | 58 (74) | 0.02 |
| Tooth brushing (Y) | 12 months | | 93 (70) | 42 (69) | 51 (70) | 0.90 |
| | 18 months | | 119 (94) | 56 (93) | 63 (94) | 0.87 |
| | 24 months | | 117 (73) | 56 (100) | 61 (98) | 0.34 |
| Oral thrush diagnosis (Y) | | | 23 (14) | 3 (4) | 20 (26) | **<0.0001** |
| Diaper rash diagnosis (Y) | | | 13 (8) | 6 (7) | 7 (9) | 0.70 |
| Child antibiotic treatment* (Y) | | | 29 (18) | 14 (17) | 15 (19) | 0.72 |
| Child antifungal treatment* (Y) | | | 31 (19) | 8 (10) | 23 (29) | **0.002** |
| 2y Child *C. albicans* detection (Y) | | | 78 (49) | na | na | na |
| 2y Child *S. mutans* detection (Y) | | | 98 (61) | 41 (50) | 57 (73) | **0.003** |

*(Continued)*

**Table 1.** (Continued)

| Categories | | | | Child *C. albicans* Status | | |
|---|---|---|---|---|---|---|
| | | | Enrolled n=160 (%) | Negative n=82 (%) | Positive n=78 (%) | p value |
| **Maternal factors** | Age of mothers (years) | | 27.5±5.4 | 28.5±5.8 | 26.4±4.8 | **0.02** |
| | Employment (Y) | | 81 (51) | 47 (57) | 34 (44) | 0.08 |
| | Emotional condition (Y) | | 57 (36) | 26 (32) | 31 (40) | 0.29 |
| | Smoking (Y) | | 26 (16) | 15 (18) | 11 (14) | 0.47 |
| | Diabetes (Y) | | 10 (6) | 9 (11) | 1 (1) | **0.01** |
| | Hypertension (Y) | | 20 (13) | 11 (13) | 9 (12) | 0.72 |
| | Asthma (Y) | | 22 (14) | 15 (18) | 7 (9) | 0.09 |
| | Vaginal candidiasis during pregnancy (Y) | | 26 (16) | 10 (12) | 16 (21) | 0.15 |
| | Vaginal candidiasis 6m postpartum (Y) | | 2 (1) | 1 (1) | 1 (1) | 0.97 |
| | Antifungal used during pregnancy (Y) | | 38 (24) | 18 (22) | 20 (26) | 0.58 |
| | Antifungal used 6m postpartum (Y) | | 20 (13) | 11 (13) | 9 (12) | 0.72 |
| | Antibiotics used during pregnancy (Y) | | 55 (34) | 22 (27) | 33 (42) | **0.04** |
| | Antibiotics used 6m postpartum (Y) | | 29 (18) | 14 (17) | 15 (19) | 0.72 |
| | Marital status (Married) | | 35 (22) | 23 (28) | 12 (15) | 0.053 |
| | Number of children | 0 | 53 (33) | 28 (34) | 25 (32) | 0.92 |
| | | 1 | 45 (28) | 22(27) | 23 (30) | |
| | | ≥2 | 62 (39) | 32 (39) | 30 (38) | |
| | Education | Middle school | 15 (9) | 7 (9) | 8 (10) | 0.40 |
| | | High school | 83 (52) | 38 (46) | 45 (58) | |
| | | Associate | 21 (13) | 13 (16) | 8 (10) | |
| | | ≥College | 41 (26) | 24 (29) | 17 (22) | |
| | Tooth brushing | Not daily | 11 (7) | 4 (5) | 7 (9) | 0.33 |
| | | Once | 46 (29) | 21 (26) | 25 (32) | |
| | | Twice | 103 (64) | 57 (69) | 46 (59) | |
| | Salivary *C. albicans* | No | 79 (49) | 50 (61) | 29 (37) | **0.002** |
| | | 1-400CFU/ml | 38 (24) | 19 (23) | 19 (24) | |
| | | >400CFU/ml | 43 (27) | 13 (16) | 30 (38) | |
| | Plaque *C. albicans* | No | 102 (64) | 58 (71) | 44 (57) | **0.16** |
| | | 1-400CFU/ml | 14 (9) | 7 (8) | 7 (9) | |
| | | >400CFU/ml | 43 (27) | 17 (21) | 26 (34) | |
| | Salivary *S. mutans* | $<10^5$ CFU/ml | 54 (34) | 29 (35) | 25 (32) | 0.66 |
| | | $\geq10^5$ CFU/ml | 106 (66) | 53 (65) | 53 (68) | |
| | Plaque *S. mutans* | $<10^6$ CFU/ml | 102 (64) | 53 (65) | 49 (64) | 0.90 |
| | | $\geq10^6$ CFU/ml | 57 (36) | 29 (35) | 28 (36) | |
| | Decayed teeth number | | 2.7±3.8 | 2.4±3.4 | 3.1±4.1 | 0.25 |
| | Decayed teeth | ≤3 | 116 (73) | 64 (78) | 52 (67) | 0.11 |
| | | >3 | 44 (27) | 18 (22) | 26 (33) | |
| | Missing teeth number | | 1.1±2.8 | 1.0±1.8 | 1.2±3.5 | 0.69 |
| | Missed teeth | ≤3 | 148 (92) | 77 (94) | 71 (91) | 0.49 |
| | | >3 | 12 (8) | 5 (6) | 7 (9) | |
| | Filled teeth number | | 3.0±3.4 | 3.3±3.4 | 2.7±3.4 | 0.31 |
| | Filled teeth | ≤3 | 103 (64) | 52 (63) | 51 (65) | 0.79 |
| | | >3 | 57 (36) | 30 (37) | 27 (35) | |

*Includes either IV, oral suspension, cream/ointments/suppository, or shampoo.

Results are shown in prevalence n (%) or mean (standard deviation).

the age of two years (p-value = 0.002 and 0.003 respectively). With respect to maternal characteristics, children with *C. albicans* detections were born to mothers with lower mean ages (p-value = 0.02), free of diabetes (p-value = 0.01), and who had received antibiotics treatment during pregnancy (p-value = 0.04), in comparison to those in the *C. albicans* negative group. Moreover, children whose mothers had higher salivary *C. albicans* carriage levels had a significantly higher rate of *C. albicans* detection (p-value = 0.002). No significant differences were observed in relation to mothers' educational levels, which encompassed middle school, high school, an associate degree typically completed in two years, a bachelor's degree typically requiring four years, and >College, which is post-graduate study. S2 Fig displays the various feeding practices, such as exclusive breastfeeding, exclusive bottle feeding, a combination of both, night breastfeeding, and night bottle feeding among the enrolled children. At two months, night bottle feeding was significantly higher in the *C. albicans* positive group (p = 0.02).

### *Candida* transmission profile in mother-child dyads

As shown in Fig 1, we compared the detection of *Candida spp.* in mothers and children at various study time points. The proportion of mother-child pairs with positive *Candida* detection increased from 9% at one month to 36% at 12 months, then stabilized at 33% from 12 to 18 months before slightly decreasing to 29% at 24 months, as depicted in the upper panel of Fig 1. The middle panel of the figure shows the percentage of mothers and children sharing the same type (green) or different types (blue) of *Candida spp.* Overall a total of 36 mother-child dyads (22.5% of all pairs) had simultaneous colonization by *C. albicans*, three (1.9% of all pairs) by *C.*

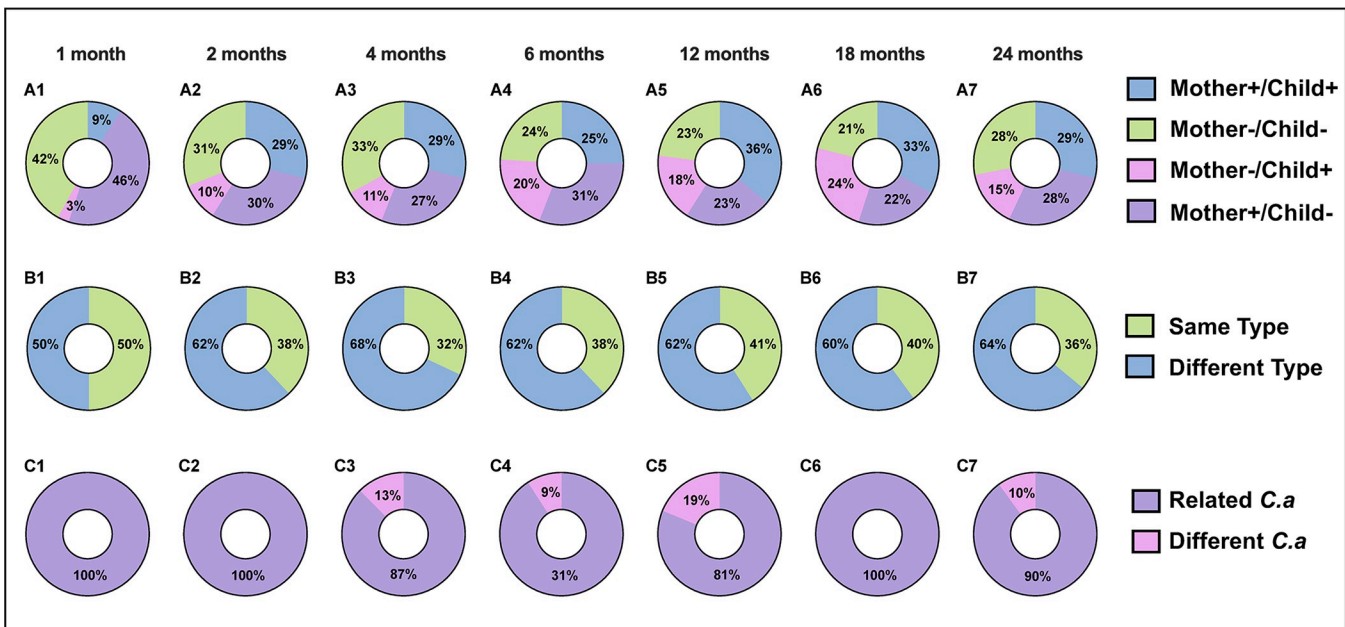

**Fig 1. Oral *Candida* comparison among mother-child dyads in the first two years of child's life.** The upper panel demonstrates oral *Candida* detection status among mother-child dyads. The status is categorized as mother and child both positive of *Candida* (blue), mother and child both negative of *Candida* (green), mother negative but child positive of *Candida* (pink), and mother positive but child negative (purple). Nine percent of the 36 mother-infant dyads had positive *Candida* detection in their oral cavity at 1 month, whereas 36% of the mother-infant dyads had positive *Candida* detection at 12 months. The middle panel represents the percentage of mothers and children who share the same type (green) or different type (blue) of *Candida* species, including *C. albicans*, *C. parapsilosis*, *C. dubliniensis*, *C. lusitaniae*, *C. krusei*, *C. glabrata*, or *C. tropicalis*. The lower panel indicates the genetic relatedness of *C. albicans* between mothers and children by MLST nucleotide analysis; genetically related (purple) ≥ 99% similar and genetically different (pink) <99% similar. *C.a*, *C. albicans*.

*parapsilosis*, two by *C. dubliniensis* (1.3% of all pairs), and one by *Candida lusitaniae* (0.6% of all pairs). The lower panel displays the genetic similarity of *C. albicans* between mothers and children using MLST nucleotide analysis. The purple portion represents a genetic similarity of ≥99%, indicating genetically related isolates, while the pink portion represents a genetic similarity of <99%, indicating a distinct genetic profile. Strikingly, the similarity of genotypic profiles was observed in 94% (n = 34/36) of mother-child dyads with *C. albicans*.

## Transmission rate of *C. albicans* and factors associated with vertical and horizontal transmission

The transmission of *C. albicans* was classified as vertical when children were born to mothers who tested positive for this fungus. On the other hand, horizontal transmission was used to describe cases where children were born to mothers who tested negative for *C. albicans*. To identify the factors associated with vertical and horizontal transmission of *C. albicans*, logistic regression models were utilized. These models considered various predictor variables such as demographics, delivery method, presence of siblings, attendance at daycare, mother's plaque index, mother's salivary *C. albicans* level, and feeding practices. At one month, 60% of the infants showed vertical transmission of *C. albicans*, whereas at 18 months, the percentage of vertical transmission was 36% (Fig 2A). The logistic regression analysis revealed that the mother's plaque index was significantly associated with *C. albicans* vertical transmission (OR, 8.02 [95% CI, 1.21-53.24]; p = 0.03) (Fig 2B). Additionally, Black race (OR, 3.46 [95% CI, 1.43-8.85]; p = 0.007) and daycare attendance (OR, 2.90 [95% CI, 1.18-7.39]; p = 0.02) were positively associated with an increased risk of *C. albicans* horizontal transmission (Fig 2C).

## Statistics of *C. albicans* MLST

For MLST analysis, only one sample per participant was chosen (either saliva or, if the saliva sample was negative, plaque). This resulted in a total of 227 isolates, consisting of 218 saliva samples and 9 plaque samples. A total of 215 unique DSTs were detected from the isolates, and 12 were shared by two isolates. Of the seven fragments analyzed, *VPS13* exhibited the highest discriminatory power, with 31 different allele sequences. *ZWF1b* had the second-highest discriminatory power, with 27 alleles, followed by *AAT1a* (22 alleles), *SYA1* (21 alleles), *ACC1* and *ADP1* (each with 15 alleles), and *MPIb* with the lowest variability (7 alleles). Details of the alleles and DSTs for each isolate are shown as metadata in S3 Fig, together with the age, sex, dental caries diagnosis, and source of the isolates.

## Phylogenetic analysis and global population structure of 3227 *C. albicans* isolates

The MLST data can be utilized for phylogenetic analysis using various methods with different levels of complexity. To identify robust subspecific clades, we used a variety of techniques, with a particular emphasis on techniques that rely on differences at nucleotide sites and gross allele sequence differences.

In the eBURST analysis of the 227 *C. albicans* isolates, 196 were grouped into 16 clonal clusters (CC) using a cluster definition of sharing five or more alleles (Fig 3). The largest cluster, CC 1, included 117 isolates (51.5% of all isolates), with 108 DSTs, and had DST 3883 as the putative ancestral type. CC 2 was the second largest, with 26 isolates (23 DSTs) and no putative founding DST. CC 8 was the third largest, comprising 19 isolates (19 DSTs), with type 3983 predicted as the founding type. CC 7 was composed of five isolates (5 DSTs), with DST 3923 as the putative ancestral type. The remaining CC contained at most four isolates. The remaining

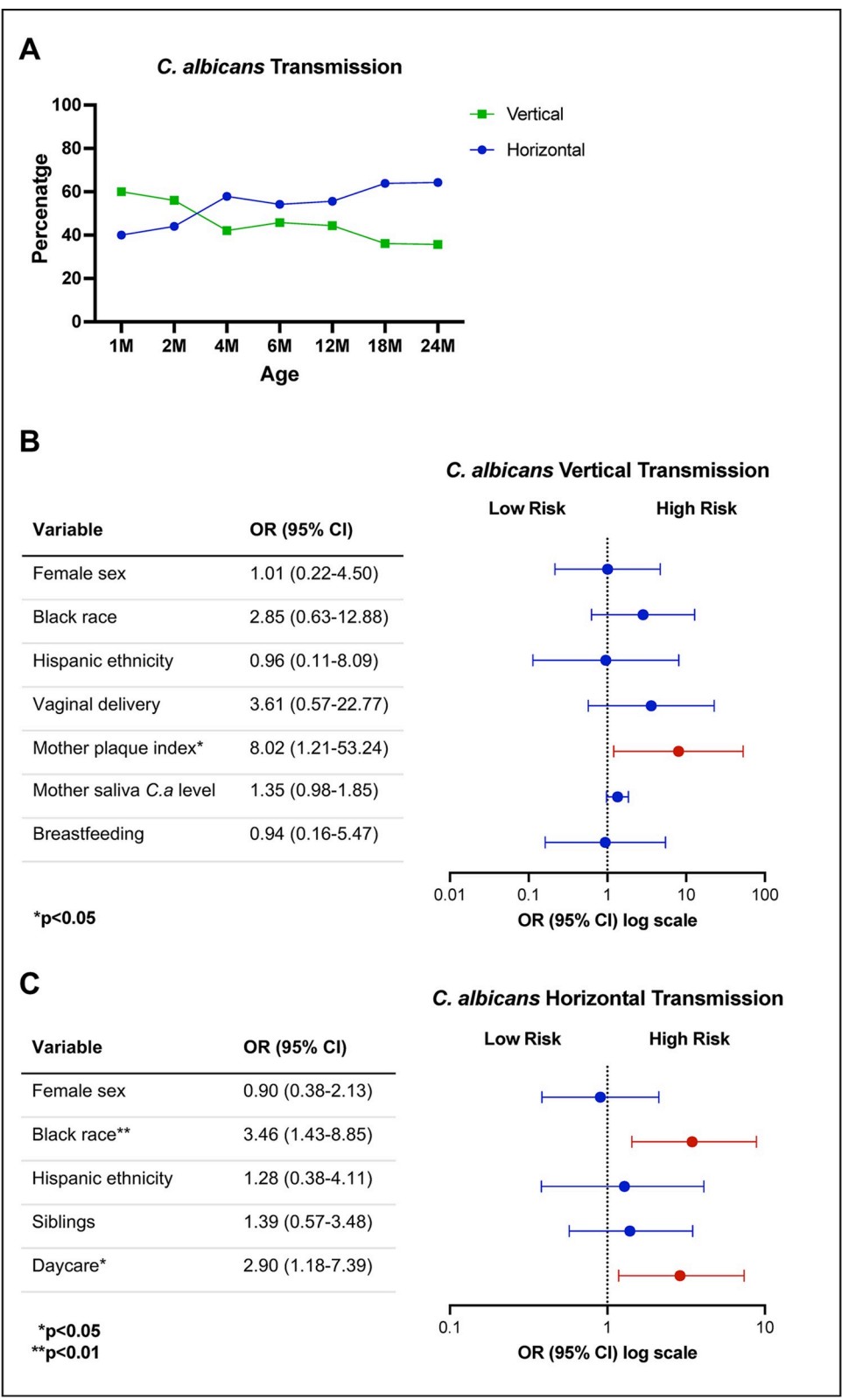

**Fig 2. Time series transmission rate of *C. albicans* and the factors associated with its vertical and horizontal transmission.** At one month, 60% of the infants displayed vertical transmission of *C. albicans*, with the remaining 40% experiencing horizontal transmission. However, by the 18-month visit, the percentage of vertical transmission was

36%, while horizontal transmission was 64%. From the multivariate logistic model, the mother's plaque index was identified as the significant factor associated with *C. albicans'* vertical transmission (OR, 8.02 [95% CI, 1.21-53.24]; p = 0.03) (B). Regarding *C. albicans* horizontal transmission, the factors associated with increased risk included being of Black race (OR, 3.46 [95% CI, 1.43-8.85]; p = 0.007) and attending daycare (OR, 2.90 [95% CI, 1.18-7.39]; p = 0.02) (C).

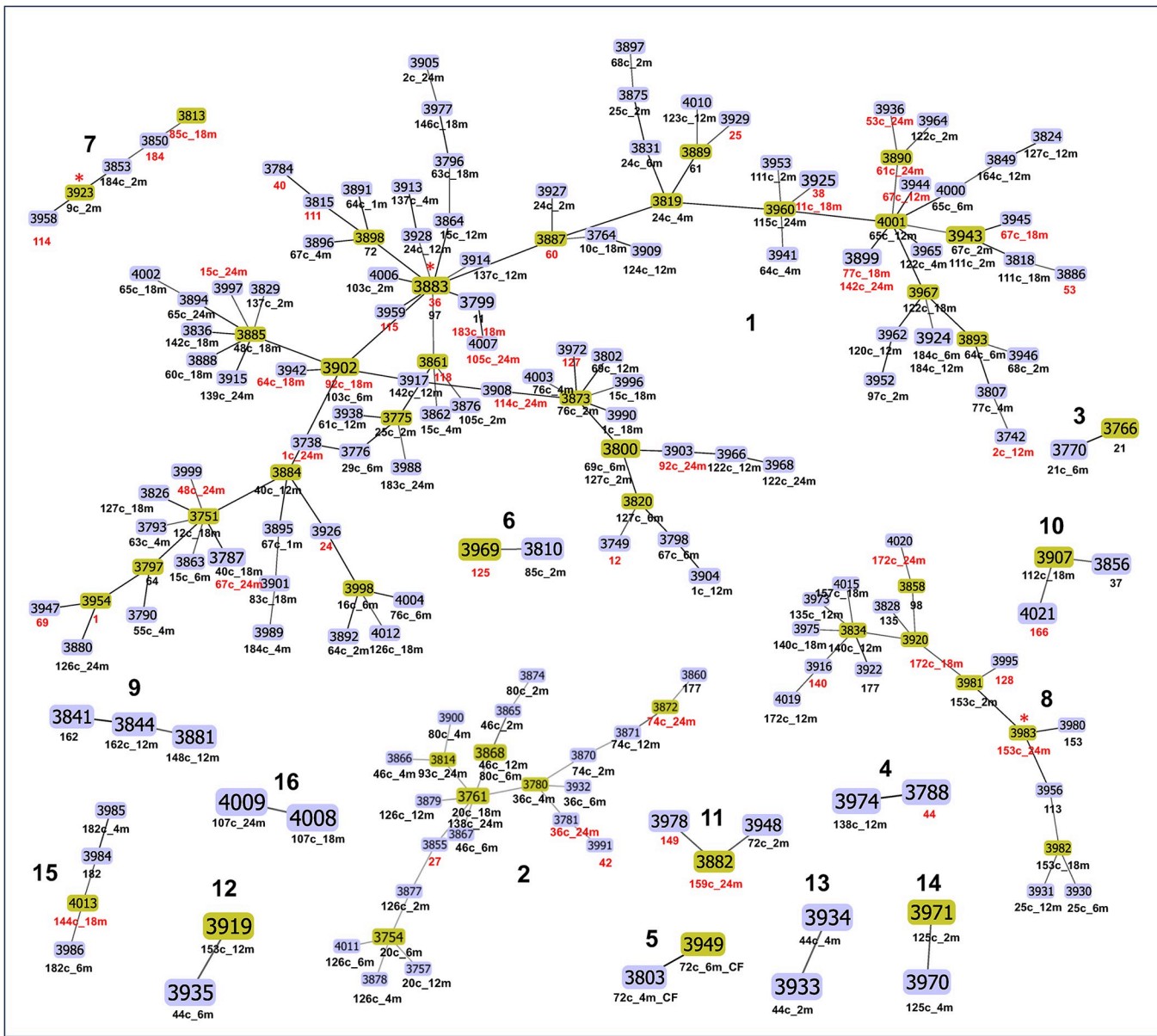

**Fig 3. Snapshot of the 16 clonal clusters identified by eBURST in 227 clinical oral isolates of *C. albicans*.** The isolates were grouped into 16 clusters and 31 singletons (not shown in the figure). Lines connecting the sequence types (DSTs) indicate a hierarchical relationship between them, with those differing in just two of the seven sequenced fragments being joined. When a putative ancestral DST was identified for a cluster, its number was marked with a red asterisk. The clonal clusters of related isolates are numbered in boldface font. Note that the color of DSTs and the length of the lines connecting the DSTs does not hold any significance. Each DST is accompanied by the subject ID and the child's age in months. If no age is specified, it signifies a mother isolate. The caries status is visually represented by font color, with black indicating no caries and red indicating the presence of caries.

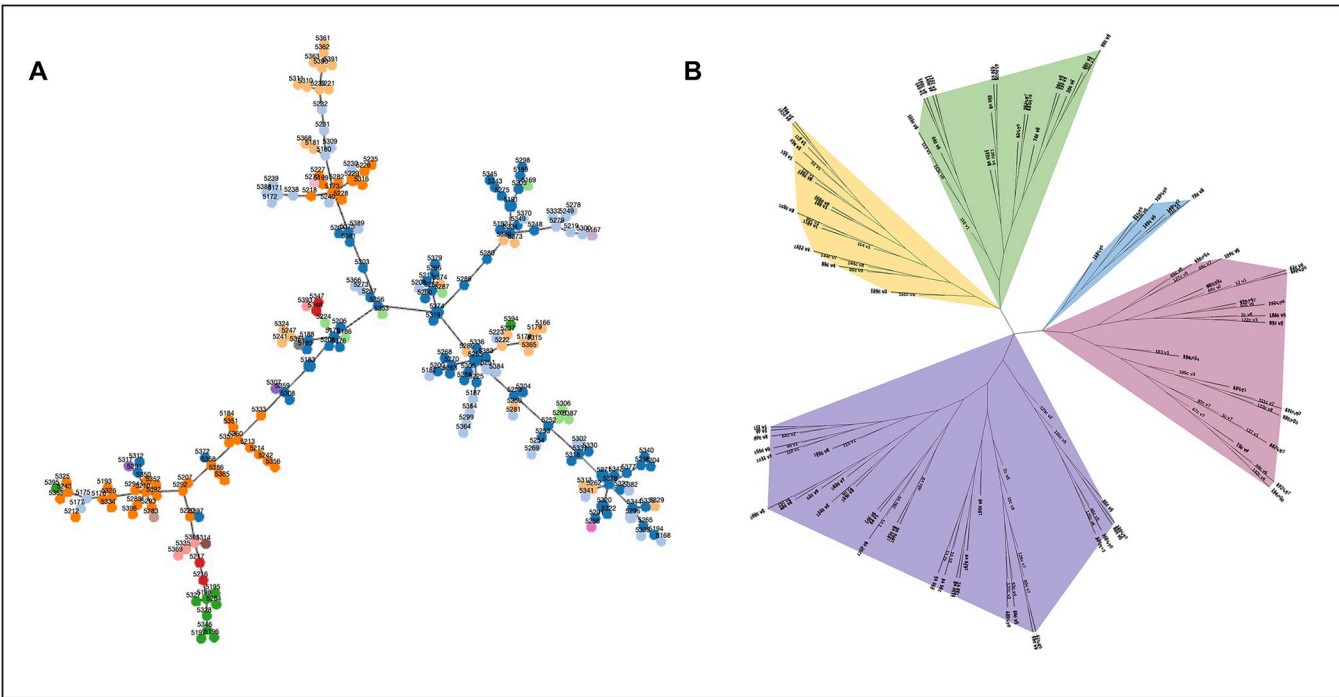

**Fig 4. Minimum spanning (MS) tree and Neighbor-Joining (NJ) tree of 227 *C. albicans* clinical oral isolates from our study.** MS tree was generated based on the allelic profiles of the 227 *C. albicans* strains isolated from our study by PHYLOViZ visualization (A). NJ tree as determined by p-distance of the 227 *C. albicans* strains isolated from our study (B). The tree is presented in a radial format to illustrate the relative positions of the isolates that are related to each other. Clades are highlighted with colored shading to help with visualization.

31 isolates (13.7%) were classified as singletons. Among pairs with *C. albicans*, the CC was identical between mother and child isolates in 64% of cases. Furthermore, we generated MS and NJ trees; these trees provide additional insights into the genetic relatedness and evolutionary history of the *C. albicans* isolates analyzed in this study (Fig 4). The NJ tree generated five closely related clades.

While the eBURST analysis provided a certain perspective on the relationships between isolates, the UPGMA pairwise-difference dendrogram generated for the 227 isolates including a reference strain of *C. albicans SC5314* (S3 Fig) took a different approach. It did not consider the specific gene fragment where sequence polymorphisms occurred, resulting in a new perspective. In the UPGMA analysis, isolates with only a few differences in the concatenated sequences across all fragments sequenced were considered similar, regardless of where the polymorphic sites were located. The UPGMA dendrogram showed that several eBURST singleton isolates were dispersed within a wide range of UPGMA clades, and isolates from different CC were mixed with one another and with several eBURST singletons.

To investigate the evolutionary relationship between DSTs, we used the first 3000 DSTs from the database (https://pubmlst.org/) to generate another UPGMA dendrogram (S4 Fig). This dendrogram classified the DSTs into 20 clades, including singletons. Of these clades, 19 had been previously identified [40], and a new clade was formed that included 16 of the study isolates. The study isolates were categorized into six clades. The largest clade (clade 1) contained 122 isolates (53.7%), while the remaining isolates were distributed among clades 2, 4, 8, 12, and the new clade. In 72% of dyads with *C. albicans*, the UPGMA clade was identical between the mother and child isolates. Additionally, the study found that isolates associated with dental caries diagnosis in mothers or S-ECC in children aged one year or older were

present in all six clades, with no significant predominance in any particular clade. Furthermore, there was no significant association observed between clade assignment and caries diagnosis.

## Discussion

S-ECC is a growing problem worldwide, particularly among low-income populations [47], and *C. albicans* has been found to be a common pathogen linked to the emergence of *S. mutans* bacteria [33] and S-ECC development. *Candida* colonization in children can result from vertical transmission from the mother or horizontal acquisition from non-maternal sources, and early colonization may contribute to a higher risk of disease later. We employed a molecular-based typing method to investigate the genetic relatedness of oral *C. albicans* between mothers and their children. Our study found that mothers can pass on highly related strains of *C. albicans* to their children during early childhood. Furthermore, *C. albicans'* vertical transmission from mothers was associated with maternal plaque index. Moreover, demographic factors such as race and daycare attendance played a role in the horizontal transmission of *C. albicans*. This highlights the crucial role of maternal socioeconomic and oral health behavior in preventing oral *Candida* colonization.

Various factors can influence *Candida* colonization, including age, population demographic, oral hygiene status, sampling techniques, and geographical variations, as well as host immunosuppression, systemic diseases like diabetes mellitus, different blood groupings, and tobacco smoking [15, 48–51]. Despite previous research indicating that individuals with diabetes are at an increased risk of *Candida* colonization due to reduced salivary flow rate compared to non-diabetic individuals [52–54], our study found that children born to diabetic mothers had a lower risk of *C. albicans* acquisition. However, in our study, we found that the Black race was the only significant factor associated with horizontal transmission of *C. albicans* among the factors mentioned above. Some studies suggested that Black individuals could exhibit distinct patterns of oral colonization by this yeast. While it is evident that individuals of Black and African American racial groups face an elevated risk of superficial and invasive *Candida* infections, the precise causal factors remain ambiguous and may be linked to underlying social determinants of health, disparities in healthcare accessibility, and various socioeconomic inequities [55]. Instead, maternal factor, such as the mother's plaque index score, was identified as a significant contributing factor to *C. albicans'* vertical transmission. Furthermore, our results showed that daycare attendance was a significant contributor to *C. albicans'* colonization. Although no studies have assessed the link between attending daycare and oral *Candida* colonization, previous research indicated that daycare increases the risk of carrying antibiotic-resistant pneumococci in children [56]. This could be attributed to the frequent use of antibiotics for upper respiratory tract infections and other unique features of daycare settings, such as children's behavior and immature immune systems [57]. Moreover, our results indicated that maternal use of antibiotics during pregnancy was associated with oral colonization by *C. albicans* in their children. Antibiotics can alter the maternal microbiota, including the vaginal and gastrointestinal environments, which might influence the risk of vertical transmission of *C. albicans* to neonates. A study by Muanda et al. (2017) suggests that maternal antibiotic use is associated with an increased risk of infant *Candida* colonization, underscoring the need for further research to elucidate the intricacies of this relationship [58].

Our study did not find a significant association between the delivery method and oral *Candida* colonization. Some authors suggest that *C. albicans* can be transmitted from mother to child, with possible sources including delivery, skin, vagina, and perianal region. In a study by Zisova et al., vaginal samples were collected from 80 mothers before delivery, and oral and

gastrointestinal samples were taken from their newborns after birth. The study found that all 13 mother-infant pairs with *C. albicans* had identical strains, suggesting vertical transmission [59].

Likewise, Al-Rusan et al. conducted a study that found the mother's vaginal *Candida* to be a significant source of oral *Candida* in newborns, especially those delivered vaginally [5]. In recent years, multiple studies have explored the transmission of *Candida* from maternal vaginal mucosa to neonates using DNA molecular typing techniques. These studies concluded that all *Candida* species present in mother-neonate pairs were indistinguishable [60, 61]. We have not collected vaginal samples that could be a source of *Candida* transmitted to the children.

Our research showed that infants in the *C. albicans* positive group were more likely to have been night bottle-fed at two months (p=0.02). In contrast, those in the *C. albicans* negative group were more likely to have been exclusively breastfed at 12 and 18 months (p=0.04, 0.02, respectively). Research on the relationship between feeding methods and *Candida* colonization in infants and children has yielded conflicting results. For example, Zöllner and Jorge found that *Candida* species were less common in predominantly breastfed infants compared to those who were bottle-fed (p < 0.05) [62]. Conversely, Darwazeh and al-Bashir studied 2-11 months infants and observed no significant differences in the frequency or density of *Candida* species between breastfed, bottle-fed, or mixed-feeding infants (p = 0.14) [49]. Similarly, a systematic review found no significant differences in oral *Candida* colonization between breastfed and bottle-fed children [63]. Further research is needed to clarify the relationship between feeding methods and *Candida* colonization in infants and children. Moreover, infants in the *C. albicans* positive group showed a higher likelihood of having a documented history of oral thrush diagnosis and previous use of antifungal medications, aligning with the results reported by Azevedo et al. [64]. Oral thrush is a common infection in early life, affecting 4%-15% of healthy children [65, 66], with *C. albicans* often being the opportunistic microorganism responsible for the infection [67].

Our study did not reveal a significant association between *Candida* colonization and having a sibling. This finding is consistent with a study by Hannula et al., where no significant association was reported between having a sibling and *Candida* colonization in a cohort of 40 infants who were followed up from 2-24 months [68]. However, other studies have reported conflicting results regarding the association between siblings and *Candida* colonization. For instance, Stecksen-Blicks et al. found that the presence of siblings increased the likelihood of oral *Candida* colonization in 12-month-old infants [65]. In contrast, Azad et al. reported a decrease in the richness and diversity of the gut microbiome in infants with older siblings [69].

In the current study, vertical transmission of *C. albicans* was significantly associated with mothers' plaque score. The oral microbiota of the child can be shaped by the mother's influence starting from gestation and continuing into early life through various pathways of vertical transmission including delivery method, close contact, feeding, and oral care practices [70–72].

MLST has proven to be an effective technique for investigating the population structure and epidemiology of *C. albicans* [39, 44, 73, 74]. Geographically, the distribution of known clades differs, with some clades being more common in certain regions of the world than others [75]. For instance, clade 2 predominates in the United Kingdom, while clades 14 and 17 are prevalent in Pacific Rim countries [75]. Clade 1, which is the largest among *C. albicans* clades, has a global distribution [74, 75]. These results imply that regional variations in the prevalence and distribution of *C. albicans* strains may have significant implications for the diagnosis and treatment of infections caused by the fungus. While McManus et al. reported a significant difference in the distribution of clades between healthy individuals and patients with periodontitis, with clade 1 being enriched in *C. albicans* isolates from periodontitis

patients [76], Gong et al. did not find a significant correlation between patients undergoing hemodialysis or healthy controls and either the sequence types or clades [40]. Similarly, our study did not show a significant correlation between clade assignment and dental caries diagnosis. Therefore, it is necessary to conduct further research on this patient population by examining a larger number of patients with dental caries over an extended period, possibly before and after treatment, and using MLST to analyze multiple *Candida* isolates retrieved from each site and investigate multiple carious lesions.

This study had some limitations that should be considered. Firstly, the study was conducted at a single center, so the generalizability of the results may be limited compared to multi-center studies [77]. Secondly, only a limited number of clinical isolates from the study participants were genotyped, mainly salivary isolates, and in cases where *C. albicans* was not detected in saliva, plaque samples were used instead. Future research could benefit from the comparison of *C. albicans* isolates obtained from various body parts (e.g., gastrointestinal and vaginal) in addition to oral samples collected from the same subjects or across different subjects. Thirdly, the sample size of mother-child pairs sharing *C. albicans* was relatively small, which limited statistical power to compare various parameters between the cases or to evaluate associations between molecular types (e.g., *C. albicans* DSTs) and clinicopathological data (e.g., gender, age, and diagnosis). Finally, while MLST is a well-established method for characterizing *C. albicans* [39, 44, 75], we did not use a complementary method to verify the genetic relatedness of *C. albicans* isolates between mothers and their children.

One strength of the study is that it is the first to use MLST to characterize *C. albicans* isolated from the oral cavities of mother-child dyads.

## Future perspectives

Future studies could use higher throughput genome-wide sequencing and more sophisticated models to better understand the population dynamics of medically relevant yeast in different epidemiological settings [78], which could inform preventative and therapeutic treatments and strategies. Moreover, future in vitro studies could help assess the virulence or cariogenic potential of those clinical *C. albicans* isolates. Additionally, it would be interesting to investigate how *C. albicans'* genotypes affect caries susceptibility in different ethnic or geographical populations.

## Conclusion

This study provides evidence supporting the transmission of highly related strains of *C. albicans* from mothers to their children, particularly during early childhood. Maternal factors, such as oral hygiene practice reflected by dental plaque accumulation were found to increase the transmission of *C. albicans* from mothers to their children. Additionally, race and daycare attendance played a significant role in the acquisition of *C. albicans* from non-maternal sources. These findings emphasize the importance of maternal socioeconomic status and oral health in preventing the transmission of oral pathogens. Incorporating screening for maternal fungal oral carriage and implementing oral health education programs during the perinatal stage may prove valuable in preventing fungal transmission in early infancy.

## Supporting information

**S1 Checklist. Inclusivity in global research.**
(DOCX)

**S1 Appendix. Additional description for methods section.**
(DOCX)

**S1 Table. Number and source of *C. albicans* isolates used in the MLST analysis.**
(DOCX)

**S2 Table. Characteristics of the seven housekeeping loci used in *C. albicans* MLST.**
(DOCX)

**S3 Table. Novel allele sequences for each locus identified in our study.**
(DOCX)

**S1 Fig. Summary of the current understanding of the ancestry and phylogeny of *Candida albicans*.** The evolutionary pathway of *C. albicans* is indicated in italicized typeface on a lighter grey background. Taxonomic classifications are indicated in plain typeface on a darker grey background. The summary was adapted from in addition to using the online databases http://www.catalogueoflife.org/ and http://www.mycobank.org/.
(DOCX)

**S2 Fig. Child feeding pattern during the first two years of life.** The proportion of children who were exclusively breastfed decreased steadily from 36% at one month to 4% at two years. Conversely, the proportion of children who were exclusively bottle-fed nearly doubled from 29% at one month to 56% at six months, then remained stable from 6 to 12 months before sharply declining at 18 months. Night breastfeeding gradually decreased from 68% at one month to 3% at the age of two years. During the first six months, approximately 70% of children were fed with a bottle at night, but this declined sharply after six months and reached 3% by the time they were 18 months old.
(DOCX)

**S3 Fig. UPGMA dendrogram of 227 *C. albicans* clinical oral isolates from our study.** 227 Isolates were grouped into 6 clades. The metadata, including the corresponding DST number, allelic profiles, eBURST clonal complex assignment, UPGMA clade, age, sex, oral source, and the diagnosis of dental caries of each isolate, is displayed on the right. A reference strain of *C. albicans SC5314* has been included.
(XLSX)

**S4 Fig. UPGMA dendrogram of 3000 *C. albicans* isolates with different DSTs from the database and 227 clinical isolates from our study.** The DSTs of the population isolates were categorized into 19 clades that were previously identified, and a new clade designated as "N". Additionally, there were some singletons present marked as S. To facilitate easy identification, we have highlighted the DSTs of the isolates from our study in green.
(XLSX)

**S1 Dataset. Metadata.**
(XLSX)

## Acknowledgments

The authors express their gratitude to the First Tooth Study members who helped in obtaining saliva and plaque samples from the study participants. They also extend their appreciation to the medical professionals, staff, and clinical administration at the University of Rochester Highland Family Medicine and Perinatal Dental Clinic for their invaluable assistance during the study visits. Furthermore, they acknowledge the University of Rochester Genomic Research Center, particularly Dr. Anthony Gaca, for performing the WGS data.

## Author Contributions

**Conceptualization:** Naemah Alkhars, Jin Xiao.

**Data curation:** Naemah Alkhars, Nisreen Al Jallad.

**Formal analysis:** Naemah Alkhars, Tong Tong Wu, Jin Xiao.

**Funding acquisition:** Jin Xiao.

**Investigation:** Naemah Alkhars.

**Methodology:** Naemah Alkhars, Nisreen Al Jallad, Jin Xiao.

**Project administration:** Naemah Alkhars.

**Software:** Naemah Alkhars.

**Supervision:** Jin Xiao.

**Validation:** Tong Tong Wu, Jin Xiao.

**Visualization:** Naemah Alkhars.

**Writing – original draft:** Naemah Alkhars, Jin Xiao.

**Writing – review & editing:** Naemah Alkhars, Nisreen Al Jallad, Tong Tong Wu, Jin Xiao.

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
