## [Decision Letter · Decision Letter 0]

14 Sep 2023

PONE-D-23-18282Multilocus sequence typing of Candida albicans oral isolates reveals high genetic relatedness of mother-child dyads in early life.PLOS ONE

Dear Dr. Xiao,

Thank you for submitting your manuscript to PLOS ONE. After careful consideration, we feel that it has merit but does not fully meet PLOS ONE’s publication criteria as it currently stands. Therefore, we invite you to submit a revised version of the manuscript that addresses the points raised during the review process.

We look forward to receiving your revised manuscript.

Kind regards,

Geelsu Hwang, Ph.D.

Academic Editor

PLOS ONE

Additional Editor Comments:

The reviewers asked revisions to be made before publication. Please check the reviewer's comments carefully and address them accordingly.

Reviewers' comments:

Reviewer's Responses to Questions

**Comments to the Author**

1. Is the manuscript technically sound, and do the data support the conclusions?

Reviewer #1: Yes

Reviewer #2: Yes

2. Has the statistical analysis been performed appropriately and rigorously? 

Reviewer #1: Yes

Reviewer #2: Yes

3. Have the authors made all data underlying the findings in their manuscript fully available?

Reviewer #1: Yes

Reviewer #2: Yes

4. Is the manuscript presented in an intelligible fashion and written in standard English?

Reviewer #1: Yes

Reviewer #2: Yes

5. Review Comments to the Author

Reviewer #1: Overview and general recommendation to the authors:

The presented manuscript, “Multilocus sequence typing of Candida albicans oral isolates reveals high genetic relatedness of mother-child dyads in early life.”, is an original paper that aimed to assess the vertical and horizontal transmission of oral Candida albicans isolates from the oral cavity of children. The authors concluded that a high number of isolates from children had highly genetically related strains. An important maternal factor associated with the vertical transmission of C. albicans was dental plaque accumulation, whereas race and daycare attendance were associated with horizontal transmission. The authors highlight the importance of the mother in the transmission of C. albicans in early life.

Overall, this manuscript is well written, the subject is innovative and of utmost importance. The title is both appropriate and informative, the abstract sums up well the manuscript and its goal. The “Introduction” provides a good overview of this topic. The “Materials and Methods” are clear. The “Results” are well organized in the text and the tables and figures complement it. However, I do have some questions about the “Results”, “Discussion”, and “Conclusions”. Please find below my concerns and comments on this manuscript. I ask that the authors to consider my suggestions and address each of my comments in their response.

Major comments:

- Although A1 Appendix contains a description of the collection methods, I would like the authors to elaborate and provide more details on the collection protocol. The location of sample collection, the moment of the day, and the requirements for the collection of the samples (e.g. if the participants had to refrain from eating or toothbrushing for a certain period of time before sample collection) should be described. Also, the authors mentioned that mothers had a clinical evaluation. However, in Table 1 (“Results”) the authors have a variable called “Severe early childhood caries”. Did the children also have an oral assessment?

- In line 171, the authors mentioned that the isolates were identified based on their specific color. However, the authors do not describe how they selected and preserved the isolates for subsequent analysis. They also do not state how many isolates they selected per sample. I would recommend the authors to elaborate on that.

- In the section “Inclusion and Exclusion Criteria”, one of the exclusion criteria was “having received oral and/or systemic antifungal treatment prior to the initial study visit.”. It was not clear when the initial study visit took place. Also, why did the authors exclude infants that took antifungals, but not infants who took antibiotics? These are also known to increase the risk of subsequent fungal infections.

- In the “Results”, the authors should write the exact p-values whenever these are <0.05. Also, did the authors ask the participants if they had taken any antibiotics in the previous month (or recently)?

- In Table 1, the amount of C. albicans is expressed in two different ways, namely “Salivary C. albicans (103 CFU/ml)” and “Salivary C. albicans” followed by categories (and the same for S. mutans). I do not understand the first way of categorizing these counts. What does “Salivary C. albicans (103 CFU/ml)” mean? I would advise choosing only one way of presenting the results, as it gets a bit confusing. For plaque samples, only one way was used to present the counts.

- In Table 1, the authors presented the “decayed teeth number” and the “missing teeth number”. However, the “filled teeth number” is missing. Given that the authors measured the DMFT, I would suggest also including this number on the table. Additionally, I would like to ask if the authors tested the association between the overall DMFT score with the presence/absence of C. albicans.

- From lines 287 to 289, the authors describe the dynamics of the colonization over time. They describe that “The group that mother and child are both positive for Candida detection continues to increase from 1 month (9%) to 12 months (36%)”. Did the authors test to check if the increase was significant? The same question applies to lines 304-206, where the authors report that “The vertical transmission displayed a continuous decrease, starting at 60% after one month, declining to 36% at 18 months, and remaining stabile by 24 months (Fig 2A).”

- After logistic regression, the authors found that the mother’s plaque index was associated with C. albicans vertical transmission. Did the authors ask if the mothers have the habit of licking the child’s pacifier, kissing the child on the lips, or sharing cutlery? I would also recommend adding to the “Discussion” a brief comment about the potential routes for vertical transmission.

- In the subchapter “Statistic of C. albicans MLST”, the authors declared to have used a total of 227 isolates for MLST. Could the authors elaborate further on how they selected the isolates and how many isolates did they select per sample? This can be added to the “Materials and Methods” section.

- In “Discussion”, please discuss the fact that you found that children from diabetic mothers were associated with the absence of C. albicans in their oral cavity, contrary to what was expected. Also, the fact that Black race was a factor associated with horizontal Candida albicans transmission should be discussed. Lastly, the authors also observed that maternal antibiotic use during pregnancy was associated with oral colonization by C. albicans in children. Could the authors elaborate on that in the “Discussion”?

- In this study, the authors did not find an association between clade assignment and dental caries diagnosis. However, clade 1 is associated with some oral diseases. Do the authors have any information about the virulence or cariogenic potential of this or the other clades? Would it be useful to perform in vitro experiments to assess the cariogenic potential of these isolates? Maybe this can be added in a paragraph about future perspectives.

- In the “Conclusions” and “Abstract”, the authors suggest implementing strategies to reduce Candida albicans levels in mothers at risk by using antifungals. I disagree with the authors regarding this suggestion. The use of antifungals has an impact on the overall oral mycobiome (besides potentially having an impact also in the gut mycobiome, if the antifungal is taken orally). The oral mycobiome is not composed exclusively by C. albicans and it has an important role in maintaining oral homeostasis. Moreover, nowadays we face a real worldwide problem with antifungal resistance, and thus antibiotics should not be used as a preventive measure. Besides, dental caries is a complex disease associated with biofilms and not a single species. In Table 1, the authors did not observe differences in the presence of Candida albicans in children whose mothers took antifungals during pregnancy. Lastly, the authors verified that the levels of Candida albicans in children were correlated with maternal dental plaque. Given the results from this study, I would strongly suggest rephrasing the “Conclusions” and “Abstract”. Alternative strategies may be more useful and less harmful to prevent the transmission of C. albicans vertically other than the use of antifungals (e.g. oral health education programs during pregnancy, which represents a period of time when parents are particularly available to learn).

Minor comments:

- A reference is missing on line 70 (“This early colonization poses a significant health risk to immunocompromised infants, particularly those with low birth weights.”).

- Please split the sentence that goes from lines 90 to 98 into smaller sentences, as it is difficult to read.

- In Table 1, there may be some typos on the line describing the variable “Salivary C. albicans (103 CFU/ml) 2.8±18.2 .10±.49 5.6±25.8 .065”. Also, in Table 1 please clarify the difference between “Associate” and “>College” in the “Education” variable.

- In line 432 of the “Discussion”, the authors mention a “systematic review and meta-analysis”, but the paper referenced is only a systematic review. Please revise the reference.

Reviewer #2: The authors of this study employ a molecular-based typing method to elucidate the phylogenetic relationship between oral Candida samples from mothers and their respective children. The endeavor to understand the vertical/horizontal transmission of C. albicans is timely and pertinent. This topic has noteworthy implications for both microbiological research and public health. However, while the research is grounded in a solid premise, there are areas that demand clearer explication or deeper exploration.

Major Comments:

1. The methodology section is unclear on the number of clinicians involved in oral examinations and sample collection. If multiple clinicians were involved, was there a standardization process to mitigate inter-clinician variability? This is crucial to ensure the consistency and reliability of the collected samples.

2. The samples were procured from both dental plaque and saliva. These niches might harbor different microbial communities; thus, it's crucial to segregate data derived from these sources. Furthermore, in Table 1, the absence of data on plaque C. albicans from maternal samples requires clarification.

3. In the phylogenetic tree analysis, the inclusion of a reference or control strain, such as SC5314, would have been advantageous. This would provide a benchmark against which the other strains could be compared.

4. The investigation would have benefited from a longitudinal design. Tracking the same mother-child dyads over a prolonged period might offer insights into the persistence, change, or evolution of the C. albicans strains.

Minor Clarifications/Edits:

1. Line261-262: there seems to be no data presented in neither Table1 or S2 Fig for the statement. Please address this.

2. Could authors explain the rationale behind not obtaining maternal samples before childbirth? Such a baseline would have added depth to the understanding of Candida transmission dynamics.

3. The health status, especially any history of salivary gland diseases in both mothers and children, could influence Candida colonization. It would be helpful if the authors provide details or exclusion criteria related to this.

4. S4 Fig, as currently presented, lacks clarity, possibly due to file resolution issues during upload. I recommend resubmitting a high-resolution version to ensure clarity for readers.

6. PLOS authors have the option to publish the peer review history of their article (what does this mean?). If published, this will include your full peer review and any attached files.

Reviewer #1: No

Reviewer #2: No

---

## [Author Response · Author response to Decision Letter 0]

24 Oct 2023

Response: We appreciate the reviewer’s positive comments!

Major comments:

1- Although A1 Appendix contains a description of the collection methods, I would like the authors to elaborate and provide more details on the collection protocol. The location of sample collection, the moment of the day, and the requirements for the collection of the samples (e.g. if the participants had to refrain from eating or toothbrushing for a certain period of time before sample collection) should be described. Also, the authors mentioned that mothers had a clinical evaluation. However, in Table 1 (“Results”) the authors have a variable called “Severe early childhood caries”. Did the children also have an oral assessment?

Response: Thank you for your comments! We have added more detailed information on clinical sample collection in the appendix.

2- In line 171, the authors mentioned that the isolates were identified based on their specific color. However, the authors do not describe how they selected and preserved the isolates for subsequent analysis. They also do not state how many isolates they selected per sample. I would recommend the authors to elaborate on that.

Response: Thank you for your comments! More details were added. “Two isolates per sample were selected and stored in a sterilized 1.5 ml Eppendorf tube and kept frozen in -80 °C freezer for future use. For our current study, one isolate was used for subsequent analysis”. Lines 174-176.

3- In the section “Inclusion and Exclusion Criteria”, one of the exclusion criteria was “having received oral and/or systemic antifungal treatment prior to the initial study visit.”. It was not clear when the initial study visit took place. Also, why did the authors exclude infants that took antifungals, but not infants who took antibiotics? These are also known to increase the risk of subsequent fungal infections.

Response: We appreciate your comments! A new paragraph was added to the methods section, which explained the study visits (Lines 200-207). We excluded those who took antifungal medications because it could eliminate the detection of Candida. You are certainly correct, antibiotic treatment could increase the risk of fungal infection.

4- In the “Results”, the authors should write the exact p-values whenever these are <0.05. Also, did the authors ask the participants if they had taken any antibiotics in the previous month (or recently)?

Response: Thank you for your comments! Exact p-values were written instead of < 0.05. 

Yes, all participants were asked about medication consumption including antibiotic treatment and verified with their medical records.

5- In Table 1, the amount of C. albicans is expressed in two different ways, namely “Salivary C. albicans (103 CFU/ml)” and “Salivary C. albicans” followed by categories (and the same for S. mutans). I do not understand the first way of categorizing these counts. What does “Salivary C. albicans (103 CFU/ml)” mean? I would advise choosing only one way of presenting the results, as it gets a bit confusing. For plaque samples, only one way was used to present the counts.

Response: Thank you for your comments! The first line/way represents the mean value � standard deviation, and the second way is break down of the count to different categories. For simplicity, I removed the mean values.

6- In Table 1, the authors presented the “decayed teeth number” and the “missing teeth number”. However, the “filled teeth number” is missing. Given that the authors measured the DMFT, I would suggest also including this number on the table. Additionally, I would like to ask if the authors tested the association between the overall DMFT score with the presence/absence of C. albicans.

Response: Thank you for your comments! Additional rows have been added to the table to include the association between filled teeth and C. albicans status.

Yes, we tested the association between the overall DMFT score with the presence/absence of C. albicans. A significant association was found between mothers’ DMFT and mother salivary and plaque C. albicans carriage.

7- From lines 287 to 289, the authors describe the dynamics of the colonization over time. They describe that “The group that mother and child are both positive for Candida detection continues to increase from 1 month (9%) to 12 months (36%)”. Did the authors test to check if the increase was significant? The same question applies to lines 304-206, where the authors report that “The vertical transmission displayed a continuous decrease, starting at 60% after one month, declining to 36% at 18 months, and remaining stabile by 24 months (Fig 2A).”

Response: Thank you for your comments! We revised the sentence to avoid confusion. We described the descriptive data of the percentage of mothers and infants who had positive Candida detection at different points, no statistical tests were needed. Similarly, for the transmission we looked at the percentage of infants who were colonized with Candida transmitted by the mothers. Revised sentences, “Nine percent of the 36 mother-infant dyads had positive Candida detection in their oral cavity at 1 month, whereas 36% of the mother-infant dyads had positive Candida detection at 12 months”. 

“At one month, 60% of the infants showed vertical transmission of C. albicans, whereas at 18 months, the percentage of vertical transmission was 36% (Fig 2A)”. 

“At one month, 60% of the infants displayed vertical transmission of C. albicans, with the remaining 40% experiencing horizontal transmission. However, by the 18-month visit, the percentage of vertical transmission was 36%, while horizontal transmission was 64%”. Lines 321-322; 337-339; 355-357. 

8- After logistic regression, the authors found that the mother’s plaque index was associated with C. albicans vertical transmission. Did the authors ask if the mothers have the habit of licking the child’s pacifier, kissing the child on the lips, or sharing cutlery? I would also recommend adding to the “Discussion” a brief comment about the potential routes for vertical transmission.

Response: Thank you for your comments! One of the limitations of the study is that we did not ask about the habit of licking the child’s pacifier, kissing the child on the lips, or sharing utensils that could contribute the transmission. Possible routes of vertical transmission have been added to the discussion. Lines 510-513.

9- In the subchapter “Statistic of C. albicans MLST”, the authors declared to have used a total of 227 isolates for MLST. Could the authors elaborate further on how they selected the isolates and how many isolates did they select per sample? This can be added to the “Materials and Methods” section.

Response: Thank you for your comments! A table has been created and added to the supplemental to show the number of isolates used at each study visit and their sources (Table S1). Additionally, more detailed information has been added to the materials and method regarding isolates selection. Lines 174-184.

10- In “Discussion”, please discuss the fact that you found that children from diabetic mothers were associated with the absence of C. albicans in their oral cavity, contrary to what was expected. Also, the fact that Black race was a factor associated with horizontal Candida albicans transmission should be discussed. Lastly, the authors also observed that maternal antibiotic use during pregnancy was associated with oral colonization by C. albicans in children. Could the authors elaborate on that in the “Discussion”?

Response: Thank you for your comments! Several sentences have been added to the discussion concerning diabetes, black race, and maternal use of antibiotics. Lines 449-452; 454-459; 466-472.

 “Despite previous research indicating that individuals with diabetes are at an increased risk of Candida colonization due to reduced salivary flow rate compared to non-diabetic individuals (52-54), our study found that children born to diabetic mothers had a lower risk of C. albicans acquisition. 

“Some studies suggested that Black individuals could exhibit distinct patterns of oral colonization by this yeast. While it is evident that individuals of Black and African American racial groups face an elevated risk of superficial and invasive Candida infections, the precise causal factors remain ambiguous and may be linked to underlying social determinants of health, disparities in healthcare accessibility, and various socioeconomic inequities (55)”.

“Moreover, our results indicated that maternal use of antibiotics during pregnancy was associated with oral colonization by C. albicans in their children. Antibiotics can alter the maternal microbiota, including the vaginal and gastrointestinal environments, which might influence the risk of vertical transmission of C. albicans to neonates. A study by Muanda et al. (2017) suggests that maternal antibiotic use is associated with an increased risk of infant Candida colonization, underscoring the need for further research to elucidate the intricacies of this relationship (58)”.

11- In this study, the authors did not find an association between clade assignment and dental caries diagnosis. However, clade 1 is associated with some oral diseases. Do the authors have any information about the virulence or cariogenic potential of this or the other clades? Would it be useful to perform in vitro experiments to assess the cariogenic potential of these isolates? Maybe this can be added in a paragraph about future perspectives.

Response: Thank you for your comments! Clade 1 has been reported to be associated with periodontitis. We do not have any information about the cariogenic potential of this clade. A sentence has been added to the new paragraph titled Future Perspectives, to investigate the virulence of theses C. albicans in future studies. Lines 550-551.

12- In the “Conclusions” and “Abstract”, the authors suggest implementing strategies to reduce Candida albicans levels in mothers at risk by using antifungals. I disagree with the authors regarding this suggestion. The use of antifungals has an impact on the overall oral mycobiome (besides potentially having an impact also in the gut mycobiome, if the antifungal is taken orally). The oral mycobiome is not composed exclusively by C. albicans and it has an important role in maintaining oral homeostasis. Moreover, nowadays we face a real worldwide problem with antifungal resistance, and thus antibiotics should not be used as a preventive measure. Besides, dental caries is a complex disease associated with biofilms and not a single species. In Table 1, the authors did not observe differences in the presence of Candida albicans in children whose mothers took antifungals during pregnancy. Lastly, the authors verified that the levels of Candida albicans in children were correlated with maternal dental plaque. Given the results from this study, I would strongly suggest rephrasing the “Conclusions” and “Abstract”. Alternative strategies may be more useful and less harmful to prevent the transmission of C. albicans vertically other than the use of antifungals (e.g. oral health education programs during pregnancy, which represents a period of time when parents are particularly available to learn).

Response: Thank you for your comments! The suggestion of implementing antifungal treatments have been removed and replaced with oral health education programs. Line 48 & 552.

Minor comments:

1- A reference is missing on line 70 (“This early colonization poses a significant health risk to immunocompromised infants, particularly those with low birth weights.”).

Response: Thank you for the catch. The reference has been added. Line 63.

2- Please split the sentence that goes from lines 90 to 98 into smaller sentences, as it is difficult to read.

Response: Thank you for your comment! The sentence has been split for easy understanding. Line 84-90.

3- In Table 1, there may be some typos on the line describing the variable “Salivary C. albicans (103 CFU/ml) 2.8±18.2 .10±.49 5.6±25.8 .065”. Also, in Table 1, please clarify the difference between “Associate” and “>College” in the “Education” variable.

Response: Thank you for your comment! The first reviewer asked for choosing one way to represent the variable, so we decided to remove the mean values of microbiological counts from the table.

An associate degree is typically earned after two years, while a bachelor's degree typically takes four years to complete. More than college is post-graduate study. Lines 283-286.

4- In line 432 of the “Discussion”, the authors mention a “systematic review and meta-analysis”, but the paper referenced is only a systematic review. Please revise the reference.

Response: Thank you for your comment! The sentence revised to systematic review only. Line 494.

Reviewer #2: 

The authors of this study employ a molecular-based typing method to elucidate the phylogenetic relationship between oral Candida samples from mothers and their respective children. The endeavor to understand the vertical/horizontal transmission of C. albicans is timely and pertinent. This topic has noteworthy implications for both microbiological research and public health. However, while the research is grounded in a solid premise, there are areas that demand clearer explication or deeper exploration.

Response: We appreciate the positive feedback from the reviewer!

Major Comments:

1. The methodology section is unclear on the number of clinicians involved in oral examinations and sample collection. If multiple clinicians were involved, was there a standardization process to mitigate inter-clinician variability? This is crucial to ensure the consistency and reliability of the collected samples.

Response: Thank you for your comment! A new paragraph was added to the methods sections entitled Oral Examination and Data/Sample Collection. Lines 200-207.

2. The samples were procured from both dental plaque and saliva. These niches might harbor different microbial communities; thus, it's crucial to segregate data derived from these sources. Furthermore, in Table 1, the absence of data on plaque C. albicans from maternal samples requires clarification.

Response: Thank you for your comment! Totally agree that both niches can harbor different microbial communities. As a future research direction, we are planning to segregate those niches. Mother plaque C. albicans data have been added to Table 1.

3. In the phylogenetic tree analysis, the inclusion of a reference or control strain, such as SC5314, would have been advantageous. This would provide a benchmark against which the other strains could be compared.

Response: Thank you for your comment! We have included a reference strain of SC5314 and updated the S3 Fig.

4. The investigation would have benefited from a longitudinal design. Tracking the same mother-child dyads over a prolonged period might offer insights into the persistence, change, or evolution of the C. albicans strains.

Response: Thank you for your comment! We agree with you, a longer longitudinal study will yield more valuable information. The current study ended when children were 2 years of age. We are planning to add more study visits when children get older. 

Minor Clarifications/Edits:

1. Line 261-262: there seems to be no data presented in neither Table1 or S2 Fig for the statement. Please address this.

Response: Thank you for your comment! A new row has been added to the Table1 for 2 months night bottle feeding. S2 figure shows various feeding patterns. 

2. Could authors explain the rationale behind not obtaining maternal samples before childbirth? Such a baseline would have added depth to the understanding of Candida transmission dynamics.

Response: Thank you for your comment and sorry for the confusion! More detailed information was added in the method section, “Comprehensive oral examination and data/sample collection occurred at 8 time points: prenatal (during mother’s third trimester) for the mothers”. In fact, all maternal samples were obtained before childbirth. Lines 176-178; 200-207.

3. The health status, especially any history of salivary gland diseases in both mothers and children, could influence Candida colonization. It would be helpful if the authors provide details or exclusion criteria related to this.

Response: We appreciate your comments! Medical history was carefully checked from patient report and validating using electronic medical records; and no history of salivary gland disease was detected. It is certainly a great idea to have included it as an exclusion criterion.

4. S4 Fig, as currently presented, lacks clarity, possibly due to file resolution issues during upload. I recommend resubmitting a high-resolution version to ensure clarity for readers.

Response: Thank you for your comment! A new figure has been uploaded.

---

## [Decision Letter · Decision Letter 1]

16 Nov 2023

Multilocus sequence typing of Candida albicans oral isolates reveals high genetic relatedness of mother-child dyads in early life.

PONE-D-23-18282R1

Dear Dr. Xiao,

We’re pleased to inform you that your manuscript has been judged scientifically suitable for publication and will be formally accepted for publication once it meets all outstanding technical requirements.

Kind regards,

Geelsu Hwang, Ph.D.

Academic Editor

PLOS ONE

Reviewers' comments:

Reviewer's Responses to Questions

**Comments to the Author**

1. If the authors have adequately addressed your comments raised in a previous round of review and you feel that this manuscript is now acceptable for publication, you may indicate that here to bypass the “Comments to the Author” section, enter your conflict of interest statement in the “Confidential to Editor” section, and submit your "Accept" recommendation.

Reviewer #1: All comments have been addressed

Reviewer #2: All comments have been addressed

2. Is the manuscript technically sound, and do the data support the conclusions?

Reviewer #1: Yes

Reviewer #2: Yes

3. Has the statistical analysis been performed appropriately and rigorously? 

Reviewer #1: Yes

Reviewer #2: Yes

4. Have the authors made all data underlying the findings in their manuscript fully available?

Reviewer #1: Yes

Reviewer #2: Yes

5. Is the manuscript presented in an intelligible fashion and written in standard English?

Reviewer #1: Yes

Reviewer #2: Yes

6. Review Comments to the Author

Reviewer #1: I would like to thank the authors for addressing all my previous comments. I have no further suggestions.

Reviewer #2: (No Response)

7. PLOS authors have the option to publish the peer review history of their article (what does this mean?). If published, this will include your full peer review and any attached files.

Reviewer #1: **Yes: **Maria João Maia Azevedo

Reviewer #2: No

---

## [Editor Report · Acceptance letter]

20 Nov 2023

PONE-D-23-18282R1 

Multilocus sequence typing of *Candida albicans* oral isolates reveals high genetic relatedness of mother-child dyads in early life. 

Dear Dr. Xiao:

I'm pleased to inform you that your manuscript has been deemed suitable for publication in PLOS ONE. Congratulations! Your manuscript is now with our production department. 

Kind regards, 

on behalf of

Dr. Geelsu Hwang 

Academic Editor

PLOS ONE